# Numerical Simulation Study on Spatial Diffusion Behavior of Non-Point Source Fugitive Dust under Different Enclosure Heights

**DOI:** 10.3390/ijerph20054361

**Published:** 2023-02-28

**Authors:** Jinjun Guo, Weiqi Lin, Hao Li, Zhongshan Zhang, Xiangnan Qin

**Affiliations:** 1Yellow River Laboratory, Zhengzhou University, Zhengzhou 450001, China; 2School of Water Conservancy Engineering, Zhengzhou University, Zhengzhou 450001, China; 3Heze Emergency Management Bureau, Heze 274000, China; 4China Construction Seventh Engineering Division Corp Ltd., Zhengzhou 450004, China

**Keywords:** non-point source dust, diffusion behavior, gas-solid two-phase flow, enclosure heights, air velocities

## Abstract

Non-point source fugitive dust produced during municipal road construction is one of the main ambient air pollutants gravely threatening the life and health of construction workers and residents around construction areas. In this study, a gas-solid two-phase flow model is used to simulate the diffusion behavior of non-point source dust with different enclosure heights under wind loads. Moreover, the inhibitory effect of the enclosure on the diffusion of non-point source dust from construction to residential areas is analyzed. The results show that the physical blocking and reflux effects of the enclosure can effectively restrain dust diffusion. When the enclosure height is 3–3.5 m, the concentration of particulate matter in most sections of residential areas can be reduced to less than 40 μg/m^3^. Moreover, when the wind speed is 1–5 m/s and the enclosure height is 2–3.5 m, the diffusion height of non-point source dust particles above the enclosure is concentrated in the range 1.5–2 m. This study provides a scientific basis for setting the heights of enclosures and atomization sprinklers at construction sites. Further, effective measures are proposed to reduce the impact of non-point source dust on the air environment of residential areas and health of residents.

## 1. Introduction

Municipal road construction generates considerable non-point source dust (NPSD); hence, it is categorized among the critical sources of fugitive dust in urban areas. This dust source area is wide and produces considerable amounts of pollutants. Hence, it negatively impacts the atmospheric environmental quality over a wide area and severely harms the health of residents around the construction area [1,2,3]. The diffusion behavior of NPSD is stochastic and complicated [4]. The study of its spatial diffusion characteristics and corresponding prevention and control measures is significant for improving urban air quality and protecting human health.

To effectively restrain the generation and diffusion of NPSD and minimize its impact on the residents around construction areas, researchers have studied the diffusion behavior of dust in terms of emission characteristics [5,6,7,8], chemical element characteristics [9,10], health damage assessment [2,11,12] and dust control measures [13,14,15]. Fuchs [16] firstly studied the dispersal behavior of dust systematically, and investigated the law of dust particle diffusion and dissipation, with the assumption that the dust in the air consists of aerosols. Li et al. [17] proposed a regional dust dynamics model that constructed surface dust sources and analyzed dust concentrations based on high-resolution remote sensing signals, and their results showed that urban construction activities are a critical factor in urban dust pollution. Ketchman [18] proposed an evaluation model for the life cycle of dust emissions and analyzed the results of particulate matter (PM) emissions in the excavation stage at a construction site, which verified that the main and secondary producers of dust are road transport and excavation activities, respectively. In addition to analyzing the emission concentration of dust, researchers have studied the contribution rate of dust particles with various sizes to atmospheric pollution at the microscale level. Nagendra et al. [19] studied the metal particle types and mass concentrations of PM in air, specifically PM_10_ and PM_2.5_, at different seasons, and demonstrated that the mass concentrations of PM_10_ and PM_2.5_ in winter were higher than those in summer. In addition, they found that the coarse particles in dust mainly emanate from soil, whereas fine particles mainly originate from the exhaust emissions of transport vehicles. Rai [20] measured the physical and chemical characteristics of atmospheric PM in Kanpur and revealed that the main dust sources include roads, vehicle emissions, and construction activities. Azarmi [21,22] used a differential mobility spectrometer, particle spectrometer, scanning electron microscope, and other devices to detect and analyze particles within a certain size range in construction sites and study the physical and chemical characteristics of these particles. Faber [23] used a variety of aerosol and trace gas analysis instruments to study the physical and chemical characteristics of particles emitted by earthworks and road engineering. The study revealed that construction activities dominated by earthwork were the main factors causing PM_10_ emissions in cities, and increasing soil moisture could effectively reduce dust emissions.

To reduce the impact of NPSD on the surrounding environment effectively, many scholars have investigated the factors influencing dust diffusion behavior and proposed various dust suppression measures. Chalvatzaki [24] used the ISC3 model and emission factors to study the effects of different factors on the quantity and distribution of PM emissions from municipal solid waste dumps. Wind speed, wind direction, and pile height were found to affect the generation and diffusion of PM_10_, and the concentration of PM_10_ downwind increased with the wind speed and pile height. Mueller [25] studied the dust diffusion behavior of open-air coal, soil, and aggregate piles and observed that the turbulent airflow on the surface of the piles was the mainspring for generation and diffusion of dust under low wind speed conditions, while the loose surface of piled materials, local high temperature of the pile, and obstacles that could cause eddy currents were principal triggers for increased dust concentration. In conclusion, the diffusion behavior of dust involves many factors, including wind speed, dust source state, and air humidity. Based on the characteristics of related factors, strengthening or covering the soil surface [26,27], improving soil humidity [23,28], setting up dustproof nets [29,30] and using new solidifying materials [31] are the methods mainly adopted to constrain dust generation and diffusion.

The PM in urban dust has a complex composition, wide size range, and various sources; its generation and diffusion laws are related to several factors. The dust source area of NPSD is also wide, and its diffusion, concentration and speed are closely related to the environment. Consequently, its diffusion behavior is difficult to study through field experiments. With improvements in computer performance and numerical simulation methods, an increasing number of researchers have used numerical simulation technology to study the diffusion behavior of fugitive dust [8,32,33,34,35]. Zhao [36] proposed a multi-layer coupling model system to simulate the impact of construction dust emissions on air quality for a period of four months in Beijing. Jeong [37] built a virtual wind tunnel model based on FLUENT to analyze the computational fluid dynamics (CFD) of a construction site and studied the dust concentration at the site and the dust prevention efficiency of a commercial windproof net. Hilton [38] proposed a CFD-DEM (discrete element method) coupling model in which DEM was used to simulate the motion of large particles, while CFD was applied to simulate dust and airflow, and the generation and movement of dust in the system were consequently obtained. Wang [39] established the *k-ε-θ* mathematical model based on the characteristics of gas-solid two-phase flow, and the influence of ventilation parameters on airflow and dust diffusion was simulated using FLUENT in a fully mechanized mining face. Compared with physical experiments, numerical simulation can eliminate the test limitations due to the environment, equipment, and materials. Moreover, a complex test environment can be quickly and accurately constructed, and real and effective test results can be obtained. For the simulation of dust diffusion, the gas-solid two-phase flow model [40,41] considers airflow as a continuous phase, and dust particles are regarded as a discrete phase. Compared with the Gaussian diffusion model [6,42], the foregoing model can simulate the physical diffusion state of dust particles more accurately.

In summary, municipal road construction has the characteristics of wide exposed areas of soil, large amounts of construction waste, and active transportation operations. Non-point source fugitive dust is easily produced in the process of construction and is seriously harmful to the environment and human health. Therefore, the study of the prevention and control measures of dust diffusion based on its characteristics is urgently required. It is possible to reduce the impact of dust generated in the engineering area on residential areas by setting up fences in the engineering area. However, there are still relatively few studies on the dust suppression efficiency of different enclosure heights. In consequence, in this paper, a numerical model based on the gas-solid two-phase flow model is established using the FLUENT software. The diffusion process of NPSD in the construction area under different enclosure heights and airflow velocities is investigated by numerical simulation experiments. Further, the influence of dust on residential areas under different conditions was analyzed to provide a scientific basis for the formulation and implementation of NPSD suppression measures. The remainder of this paper is organized as follows. Section 2 elaborates on the fundamental theory of the gas-solid two-phase flow model. Section 3 introduces the relevant parameters of the numerical model developed in this study. Section 4 presents the analysis of the numerical results. The diffusion behavior of NPSD is systematically examined considering different enclosure heights, and corresponding prevention and control measures are formulated.

## 2. Basic Theory of Gas-Solid Two-Phase Flow Model

### 2.1. Fundamental Equation

Gas-solid two-phase flow must conform to objective reality and the law of mechanical movement, so the gas-solid two-phase flow motion in this study must also obey the law of conservation of mass and momentum.

According to the law of conservation of mass, the mass change rate of continuous fluid in a fixed fluid region should be equal to the net flux of fluid on the surface of the region. As a result, the continuity equation can be obtained as Equation (1):(1)$$\frac{\partial \rho }{\partial t}+\nabla \cdot \rho u=0$$
where *ρ* is the density of the fluid (kg·m^−3^); and ***u*** is the velocity field of the fluid.

The municipal road construction site is an open space, and the density of air does not significantly change when it flows through a foundation pit or an enclosure; thus, the air around the site can be regarded as an incompressible fluid. Therefore, Equation (1) can be expressed as Equation (2):(2)∇⋅u=0

According to the law of conservation of momentum, the momentum change rate of a continuous fluid in a fixed fluid region should be equal to the sum of the mass force and the surface force in the region. In consequence, the momentum conservation equation is derived as Equation (3):(3)$$\rho \frac{\partial u_{i}}{\partial t}+\nabla \cdot \left(\rho u_{i}u\right)=-\frac{\partial p}{\partial x_{i}}+\frac{\partial \tau_{ii}}{\partial x_{i}}+\frac{\partial \tau_{ij}}{\partial x_{j}}+\frac{\partial \tau_{ik}}{\partial x_{k}}+\rho f_{i}$$
where p is the average pressure of the fluid element (N∙m^−2^); *τ_ij_* is the stress in the *j* direction exerted on a plane perpendicular to the *i* axis; *u_i_*, *u_j_* and *u_k_* are the velocity components of the fluid in directions *i*, *j* and *k*, respectively (m∙s^−1^); *x_i_*, *x_j_* and *x_k_* are coordinates in directions *i*, *j* and *k*, respectively (m); and *f_i_* is the body force per unit mass acting on the fluid element in direction *i*.

The stress component *τ_ij_* can be calculated by Equation (4):(4)$$\tau_{ij}=\left[\mu \left(\frac{\partial u_{i}}{\partial x_{j}}+\frac{\partial u_{j}}{\partial x_{i}}\right)\right]-\frac{2}{3}\mu \frac{\partial u_{k}}{\partial x_{k}}\delta_{ij}$$
where μ is the hydrodynamic viscosity of the fluid (N∙s∙m^−2^); and *δ_ij_* is Kronecker delta.

### 2.2. Turbulence Model

Airflow in nature is basically turbulent [43], and a two-equation model is frequently adopted in engineering simulations, with typical examples such as *k-ε* and *k-ω* models [44]. In this study, the standard *k*-*ε* model is used to establish the equations of turbulent kinetic energy *k* and the energy dissipation rate *ε* as Equations (5)–(9):(5)$$\frac{\partial \rho k}{\partial t}+\rho \nabla \cdot uk=\nabla \left[\left(\mu +\frac{\mu_{t}}{\sigma_{\varepsilon }}\right)\nabla k\right]+G_{k}+G_{b}-\rho \varepsilon -Y_{m},$$
(6)$$\frac{\partial \rho \varepsilon }{\partial t}+\rho \nabla \cdot u\varepsilon =\nabla \left[\left(\mu +\frac{\mu_{t}}{\sigma_{\varepsilon }}\right)\nabla \varepsilon \right]+\frac{\varepsilon }{k}C_{1\varepsilon }\left(G_{k}+C_{3\varepsilon }G_{b}\right)-\rho C_{2\varepsilon }\frac{\varepsilon^{2}}{k},$$
(7)$$G_{k}=\mu_{t}u_{i,j}\left(u_{j,i}+u_{i,j}\right),$$
(8)$$G_{b}=\beta g_{i}\frac{\mu_{t}}{Pr_{t}}\frac{\partial T}{\partial x_{i}},$$
(9)$$Y_{m}=2\rho \varepsilon \frac{k}{a^{2}},$$
where *G_k_* and *G_b_* are the generation terms of turbulent kinetic energy *k* caused by the mean velocity gradient and buoyancy, respectively; *β* is the coefficient of thermal expansion of fluid (K^−1^); *g_i_* is the component of gravitational acceleration in direction *i* (m·s^−2^); *μ_t_* is the turbulent viscosity coefficient (kg·m^−1^·s^−1^); *Pr_t_* is the number of Prandtl; *T* is the temperature (K); *Y_m_* is the generation term of turbulent diffusion fluctuation of compressible fluid; *a* is the coefficient of fluid thermal conductivity (m^2^·s^−1^); and *C*_1*ε*_, *C*_2*ε*_ and *C*_3*ε*_ are empirical coefficients.

For incompressible fluids, Equations (5) and (6) can be simplified to obtain Equations (10) and (11):(10)$$\rho \nabla \cdot uk=\nabla \left[\left(\mu +\frac{\mu_{t}}{\sigma_{\varepsilon }}\right)\nabla k\right]+G_{k}-\rho \varepsilon$$
(11)$$\rho \nabla \cdot u\varepsilon =\nabla \left[\left(\mu +\frac{\mu_{t}}{\sigma_{\varepsilon }}\right)\nabla \varepsilon \right]+\frac{\varepsilon }{k}C_{1\varepsilon }G_{k}-\rho C_{2\varepsilon }\frac{\varepsilon^{2}}{k}$$

### 2.3. Particle Balance Equation

The diffusion of particles in open space is a kind of gas-solid two-phase flow. As the standard *k*-*ε* model can only calculate the flow field in space, another model is needed to capture the particle diffusion. Moreover, because of the small volume fraction of the particle phase considered in this study, the discrete phase model of FLUENT is required to obtain accurate particle motion results. This model is appropriate under the condition that the volume fraction of the particle phase is less than 10%, and the mass load of the particles must exceed 10%, which agrees well with the numerical model formulated in this study. Therefore, the particle trajectory of the discrete phase can be solved by integrating the differential equation of particle force into the Laplace coordinate system. The equilibrium equation for the particle force can be expressed as Equations (12)–(14):(12)$$m_{p}\frac{du_{p}}{dt}=f_{d}+f_{g}$$
(13)$$f_{d}=\frac{1}{8}\pi d^{2}\rho_{f}C_{d}\left|u_{f}-u_{p}\right|\left(u_{f}-u_{p}\right)$$
(14)$$f_{g}=\frac{1}{6}\pi d^{3}\rho_{p}g$$
where *m_p_* is the mass of particles (kg); ***u****_p_* is the velocity of particles (m·s^−1^); ***f****_d_* is the resistance of particles; ***f****_g_* is the gravity of particles; *ρ_f_* is the density of fluid (kg·m^−3^); *ρ_p_* is the density of particles (kg·m^−3^); ***u****_f_* is the velocity of fluid (m·s^−1^); ***u****_p_* is the velocity of particles (m·s^−1^); *C_d_* is the drag coefficient; and *g* is the acceleration of gravity (m·s^−2^).

The resistance coefficient of particulate matter *C_d_* can be calculated from Equation (15):(15)Cd={24Re(1+0.15Re0.687)Re≤10000.43Re>1000

The Reynolds number Re can be calculated by Equation (16):(16)Re=ρfd|uf−up|μ

## 3. Numerical Setup

### 3.1. Model Sizes

Since a municipal road construction site is an open flow field, the higher the temporal resolution and the smaller the spatial resolution of the numerical model, the more accurate the reflection of the actual state of the flow field. However, a wider range of computing domains leads to a dramatic increase in the number of elements and the time cost of numerical calculations. To achieve a balance between computing efficiency and numerical accuracy, this study simplifies the numerical model and ignores the influence of surrounding buildings on the flow field. The flow field calculation region model is built using the SolidWorks three-dimensional modeling software, and its numerical calculation length and cross-section size are 60 m and 5 m × 12 m, respectively. The enclosure is located at the junction of the construction and residential areas and 20 m from the air inlet. The enclosure height *h* is set to 0, 2, 2.5, 3 and 3.5 m. The particles are generated over the entire range of the ground area of the construction site and spread to the residential area under airflow action. The numerical model of NPSD is shown in Figure 1a. A structured meshing scheme with hexahedral elements is adopted in this model, and the mesh is refined at the enclosure. The meshing scheme on the model is illustrated in Figure 1b.

### 3.2. Boundary Conditions

The boundary conditions of the numerical model in this study are set as follows.

(1) The left air inlet is set as the velocity inlet since the wind speed is considered as one of the influencing factors.

(2) The right exit is set as the pressure outlet because the velocity distribution at the exit boundary is difficult to quantify and the outlet velocity is also not relevant for this work.

(3) Considering that the particles are generated on the bottom of the left side of the enclosure, this wall is set as the particle entrance. The release of particles is in the form of surface jet, and the particle types are defined as inert; the surface jet source governs the mass flow rate. The particulate parameters are set according to the relevant references [45] and are listed in Table 1.

(4) The enclosure and other walls are set as non-slip walls. Their boundary conditions are shown in Figure 1a.

**Table 1 ijerph-20-04361-t001:** Parameter setting of PM.

No.	Condition	Parameter
1	Release position of PM	PM inlet
2	Minimum particle size	1.81 × 10^−7^ m
3	Average particle size	3.52 × 10^−5^ m
4	Maximum particle size	2.72 × 10^−4^ m
5	Mass flow rate	1.0 × 10^−6^ kg/s
6	Initial particle concentration	0 kg/m^3^

### 3.3. Working Condition Settings

According to the meteorological data, it is determined that the annual average wind speed in Zhengzhou is 2.86–3.38 m/s [46]. In order to investigate the diffusion law of NPSD particles under different enclosure heights, the initial horizontal wind speed of the model in this study is set to 3 m/s. Based on field surveys of several construction areas in Zhengzhou, the observed enclosure height of construction sites is from 2 to 3.2 m. Accordingly, four enclosure heights are selected for the study: 2, 2.5, 3, and 3.5 m. Non-enclosed sites are set as the control group.

## 4. Analysis of Simulation Results of Spatial Diffusion Behavior of Non-Point Source Fugitive Dust

### 4.1. State of Airflow Considering Different Enclosure Heights

The condition of airflow is an important factor affecting the release, diffusion, and settlement of dust particles. The relationship between enclosure height and PM distribution in residential areas can be analyzed by studying the influence of enclosures on airflow. In this study, a symmetric cross-section S (Figure 2) is selected as the monitoring surface of the model airflow field, and the velocity distribution on this airflow field under different enclosure heights is analyzed.

The distribution of airflow velocity in construction and residential areas with different enclosure heights is shown in Figure 3 with the initial horizontal wind speed of 3 m/s. This figure indicates that no significant changes in wind direction and speed are observed in the global scope of the model with the absence of an enclosure. When an enclosure is introduced, the characteristics of airflow field in the construction and residential areas change in three local areas. At area ➀ in front of the enclosure, the direction of airflow in the lower space changes to upward due to the presence of obstacles, and an area of updraft is formed. The NPSD particles are affected by the airflow in this area. A portion of the particles accumulate in quiet local wind areas, and the others spread to the residential area with updraft. In area ➁ above the enclosure, the lower airflow converges with the upper airflow, forming an airflow confluence area. In this area, a significant deflecting flow occurs, and the flow velocity increases due to the reduction in the flow section. In addition, the velocity of air in a small area in front of area ➁ also increases due to the influence of the confluence flow, and with the increase in the enclosure height, the range of this area also increases. After the air passes through the confluence area, the airflow area increases, and a large vortex area is formed at area ➂ on the leeward side of the enclosure. The range of the vortex zone also increases with the enclosure height. When the height of the enclosure increases to 3.5 m, the horizontal length of the vortex zone is approximately 40 m. This large-scale refluxing has a distinct negative effect on particle diffusion, reducing the concentration of dust diffusing from non-point sources to residential areas consequently.

### 4.2. Diffusion Process of NPSD Particles with Enclosure

The aforementioned analysis indicates that the airflow field inside the model forms three characteristic regions (rising, confluence, and vortex areas) when the enclosure is set between the construction and residential areas. The particle mass concentration distributions of the NPSD at different diffusion stages are examined in this section to determine the influence of these variables on particle diffusion behavior. The mass concentration distributions of dust particles at different diffusion stages are presented in Figure 4 with an initial horizontal wind speed and enclosure height of 3 m/s and 3.5 m, respectively.

As shown in Figure 4, the particle diffusion process can be divided into five stages, as follows.

Stage A represents the initial diffusion of particles. At this stage, the particles of NPSD are driven by airflow spread from the air inlet to the residential area. When the particles spread to the ascending airflow section under the combined action of the updraft and enclosure, some of the particles are deposited in the local calm area on the windward side of the enclosure, whereas other particles spread upward. The mass concentration and diffusion height of PM are high and low near and away from the construction area enclosure, respectively; moreover, no particles are detected in the residential area. In stage B, the concentration of particles on the windward side of the enclosure increases. A portion of the particles spread upward with the updraft and then spread to the airflow confluence area above the enclosure. The diffusion height increases owing to the influence of deflecting airflow. In stage C, the particles that accumulate on the windward side of the enclosure increase continuously. Concurrently, the particles across the enclosure through the airflow convergence area increase and continue to spread to residential areas. At this moment, some of the particles spread to region ➃, which leads to particle reflux under the action of the swirling airflow in the large-area vortex region. In stage D, the NPSD particles in region ➃ gradually move to the leeward side of the enclosure and accumulate under the drive of the vortex. Concurrently, the PM concentration gradually increases in the residential areas. In stage E, numerous particles accumulate on the windward and leeward sides of the enclosure. At the same time, particles are detected over a wide part of the residential areas, and the particle concentration decreases with the increase in diffusion distance.

The foregoing analysis demonstrates that the enclosure has a satisfactory inhibitory effect on the diffusion of dust particles from non-point sources. The main reasons are as follows: (1) The physical blocking effect of the enclosure can intercept the majority of the particles in the construction area, thus reducing the mass diffusion of particles from the construction area to the residential area. (2) Some of the particles that spread over the enclosure to the residential area produce reflux under the influence of eddy currents, reducing the diffusion range of particles in the residential area to a certain extent. (3) Most of the particles accumulate near the windward and leeward sides of the enclosure, hence allowing more convenient follow-up dust removal.

### 4.3. Effect of Different Enclosure Heights on Concentration Distribution of Dust Particles

To study the inhibitory effect of different enclosure heights on the diffusion behavior of NPSD particles, the diffusion behavior of these particles is discussed considering five enclosure heights when the initial horizontal wind speed is 3 m/s. As illustrated in Figure 2, the particle concentration distribution on monitoring cross-section S is observed and analyzed. The particle concentration distribution on cross-section S at different enclosure heights is plotted in Figure 5.

As shown in Figure 5, when no enclosure is set, the diffusion of the particles is uniform and smooth because the air is not distinctly disturbed in the flow process. Moreover, numerous particles migrate from the construction area to the residential area. After setting the enclosure, the concentration of particles on the windward side increases, indicating the accumulation of particles in this area. With an increase in the enclosure height, the accumulation of particles in this area gradually increases, which confirms that the number of particles blocked in the construction area also increases with the enclosure height. On the other side of the enclosure in the residential area, some of the NPSD particles reflux and accumulate on the leeward side of the enclosure under the action of eddy airflow in the vortex area. In contrast to the windward side, the concentration range of the particles in the vortex area has a negative relationship with the enclosure height. In detail, after setting the enclosure, the diffusion range of NPSD particles in the residential area increases; however, their overall concentration is lower than that without the enclosure. Moreover, with increasing enclosure height, the PM concentration gradually decreases. Specifically, when the enclosure height is 3 m, the PM only concentrate near the enclosure in the residential area; when the enclosure height is 3.5 m, the PM concentration over a wide section of the residential area is less than 40 μg/m^3^.

In summary, when no enclosure is set, numerous particles produced in the construction area smoothly spread to the residential area under the action of horizontal airflow, gravely threatening the environmental quality and human health in the residential area. The physical blocking effect of the enclosure and the influence of reflux airflow on the residential area can block the majority of the particles in the construction area and on both sides of the enclosure, effectively containing the diffusion of NPSD. With the increase in enclosure height, the diffusion distance of PM is shortened, and the concentration of PM in residential area also significantly decreases, although the diffusion height of PM increases slightly. Therefore, it is necessary to set a reasonable enclosure height, as it can reduce the accumulation of PM in residential areas. When the enclosure height is 3–3.5 m, the concentration of dust particles in most parts of the residential area can be reduced to 40 μg/m^3^, thereby meeting the particulate emission standard [47].

### 4.4. Effects of Different Wind Speeds and Enclosure Heights on Diffusion Height of Dust Particles

By setting an enclosure, the diffusion of NPSD particles from the construction area to the residential area can be effectively prevented. However, driven by airflow, a portion of the particles escape over the enclosure toward the residential area. Therefore, atomization nozzles can be installed above the enclosure to prevent further particle diffusion. Nevertheless, an atomizer with an exceedingly high spray height can result in waste of water resources, whereas a comparatively low injection height can be less effective at preventing particle diffusion. Therefore, it is of great economic and environmental advantage to investigate the diffusion height of NPSD particles over the enclosure and the optimal injection height of atomization nozzles considering different enclosure heights and wind speeds.

In this study, to investigate the diffusion law of NPSD particles in the construction area considering different wind speeds and enclosure heights, five initial horizontal wind speeds (1, 2, 3, 4, and 5 m/s) are selected considering that outdoor engineering activities are halted under strong winds and severe weather conditions. The calculation time of the numerical simulation is set to 240 s, and particle monitoring zone L shown in Figure 6 is selected. The diffusion height of NPSD particles in the monitoring zone under different enclosure heights and wind speeds is calculated for 240 s. Figure 7 describes the diffusion height distribution of NPSD particles above the enclosure under different enclosure heights and wind speeds. According to Figure 7, it can be indicated that under various working conditions, the diffusion height sample points of NPSD particles above the enclosure are fundamentally concentrated in the diffusion range 1.5–2.0 m. When the enclosure height is 2 m and the wind speed is 4 m/s, the percentage of sample points in the 1.5–2.0 m range reaches 90.42%, which is the highest proportion achieved. When the enclosure height is 3.5 m and wind speed is 1 m/s, the proportion of sample points in this range is relatively low. As shown in Figure 7a, when the initial horizontal wind speed is 1 m/s, the diffusion height of the particles increases with the enclosure height. When the enclosure height is 3.5 m, the number of samples whose particle diffusion height range is more than 2.5 m is 13, accounting for 8.75% of the samples. When the wind speed increases, the effect of the enclosure height on the particle diffusion height gradually decreases. The foregoing analysis indicates that when the wind speed is 1–5 m/s and the enclosure height is 2–3.5 m, the diffusion height of NPSD particles above the enclosure is less than 2 m. Therefore, an optimal injection height of the atomizing nozzle above the enclosure is 2 m.

### 4.5. Comprehensive Control of NPSD

During municipal road construction, all types of construction activities may generate and diffuse dust, which can jeopardize the environmental quality and residential health. Therefore, it is necessary to investigate the diffusion behavior of NPSD in municipal road construction and propose effective dust control measures. Based on the generation conditions and the numerical output of the NPSD diffusion behavior in this study, the source governance and suppression analysis of non-point dust and corresponding control suggestions are presented in this section.

(1) **Source control of NPSD**: At a construction site, filling and excavation, concrete mixing, transportation and processing of building materials, and airflow on the surface of exposed soil and construction waste dump, etc., can easily lead to the generation of NPSD particles. In this regard, three strategies including construction of roads with a hard base, sand stacking with cover, and bare land greening and covering can be adopted to reduce the interaction between the free particles on the surface and airflow, thus reducing dust production. The following specific measures can be implemented: (a) The ground at the entrance and exit of the construction site is built with a hard base to reduce the amount of loose soil and the production of dust particles during material transport. (b) Bulk materials, such as sand and gravel mounds, are concentrated, classified, and covered or sprinkled to prevent dust formation. (c) Temporary wastes, such as construction waste, must be cleared and transported immediately. A temporary dump site can also be set up at the construction site where dust control measures (such as water sprinkling and use of dustproof nets as cover) must be implemented. (d) Exposed ground sites and mounds or slopes at the construction site can be covered with dustproof nets.

(2) **External interception of NPSD**: The erection of construction site enclosures and water sprinkling of these enclosures can intercept dust particles during diffusion, as shown in Figure 8. It was observed in Section 4.3 that when the enclosure height is 3~3.5 m, the dust concentration in most parts of the residential area can be reduced to 40 μg/m^3^. The concentration of particulate matter meets the emission standard. Therefore, the enclosure height of construction sites near main road sections and residential areas in the city must exceed 3 m, while the enclosure height of construction sites on secondary areas can be appropriately reduced. Moreover, an automatic atomization nozzle is set above the enclosure with the injection height set above 2 m to intercept particles escaping from the construction. Furthermore, an automatic information collection module can also be used to monitor the wind speed and concentration of dust particles in the construction area and regulate the operation of the atomization nozzle automatically. In addition, the area surrounding the construction enclosure must be regularly cleaned to prevent the accumulated particles on both sides of the enclosure that could subsequently rise and spread.

## 5. Conclusions

In this study, a NPSD diffusion model based on the gas-solid two-phase flow model is established using the FLUENT software. The influence of different enclosure heights on the dust diffusion law of non-point sources is numerically simulated. Based on the numerical results, the following conclusions are drawn.

(1) The construction enclosure has a significant influence on the airflow in construction and residential areas. When there is no enclosure, the airflow is continuous and uniform in the global range of the model. Moreover, wind speed and direction do not change significantly. With the enclosure, the construction and residential areas generate three types of characteristic flow fields including rising, confluence, and vortex areas. Additionally, the influence range of the eddy backflow region increases in these areas with the height of the enclosure.

(2) The physical blocking effect of the construction enclosure leads to three characteristic airflow areas, which have a significant impact on the diffusion behavior of dust from non-point sources. When the enclosure is not erected, the diffusion of NPSD particles has no distinct fluctuation, and numerous particles migrate from the construction area to the residential area. After setting the enclosure, numerous particles accumulate on both sides of the enclosure owing to its blocking and eddy effects. In addition, with an increase in enclosure height, the diffusion range of dust particles in the residential area increases, and the particle concentration decreases. When the enclosure height is 3 m, the PM concentration in most of the residential area is reduced to 40 μg/m^3^.

(3) Based on the generators of NPSD particles and the numerical output of particle diffusion behavior, this study proposes strategies for the prevention and control of NPSD considering source control and external interception. The generation and diffusion of NPSD particles can be suppressed from the source by constructing roads with a hard base, covering the sand yard, and greening and covering exposed soil. Furthermore, construction enclosures can be used to close the construction area, and atomization nozzles can be installed above the enclosure to intercept the NPSD, thereby reducing the diffusion of dust particles toward residential areas.

## Figures and Tables

**Figure 1 ijerph-20-04361-f001:**
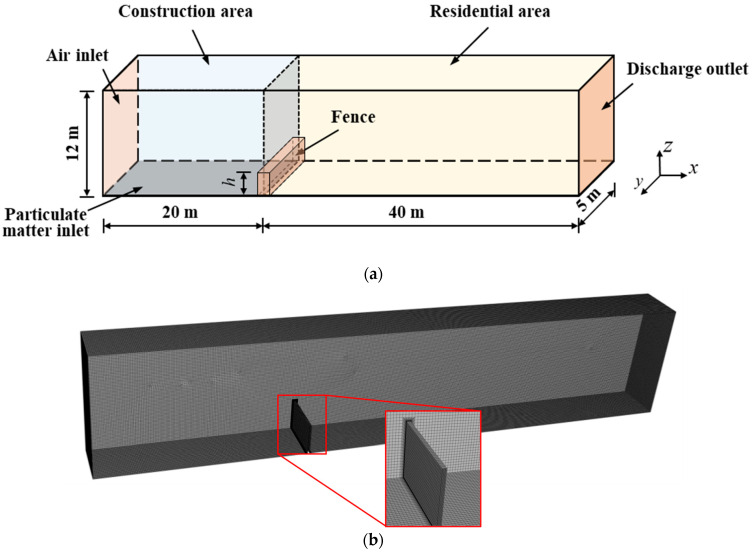
Numerical model of NPSD. (**a**) Numerical simulation model of diffusion behavior of NPSD. (**b**) Meshing effect on the numerical model.

**Figure 2 ijerph-20-04361-f002:**
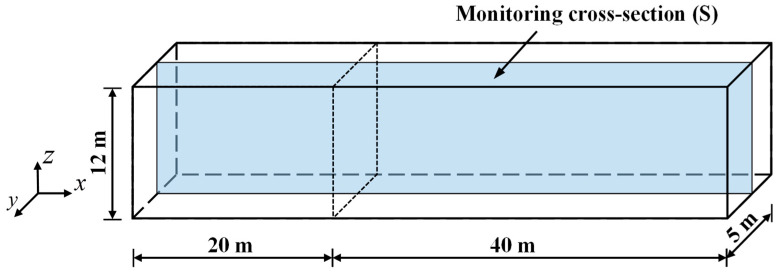
Monitoring cross-section (S) of NPSD diffusion model.

**Figure 3 ijerph-20-04361-f003:**
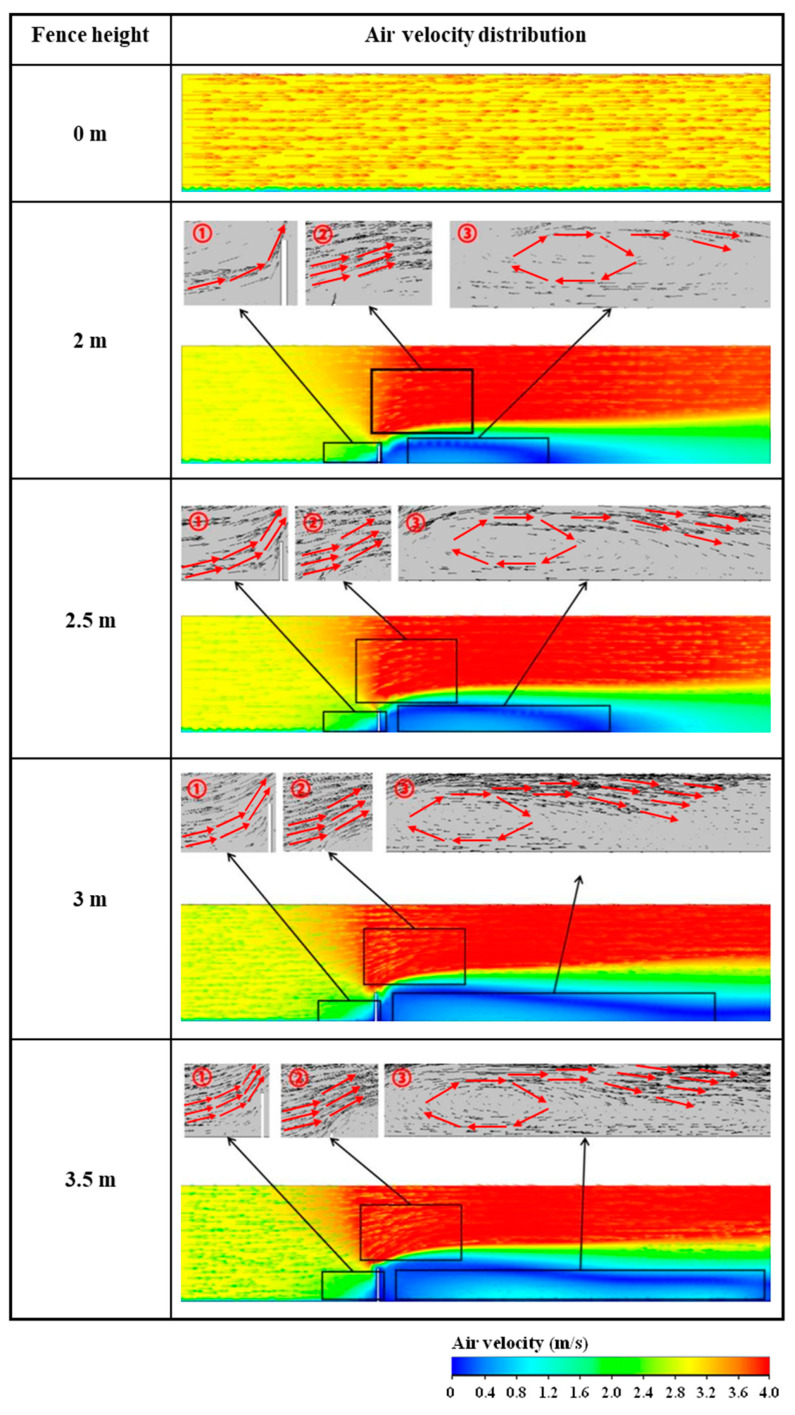
Distribution of airflow field velocity on construction and residential areas considering monitoring section S with five enclosure heights when initial horizontal wind speed is 3 m/s. ➀ is the updraft area; ➁ is the confluence area; ➂ is the vortex area.

**Figure 4 ijerph-20-04361-f004:**
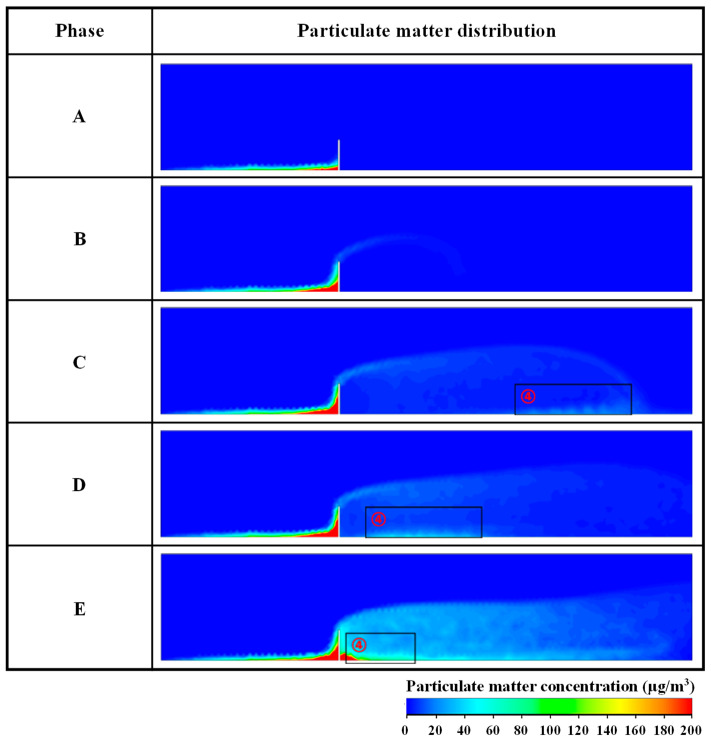
Distribution of particle mass concentration at different stages on monitoring section S when initial horizontal wind speed is 3 m/s and enclosure height is 3.5 m. ➃ is the reflux area of PM.

**Figure 5 ijerph-20-04361-f005:**
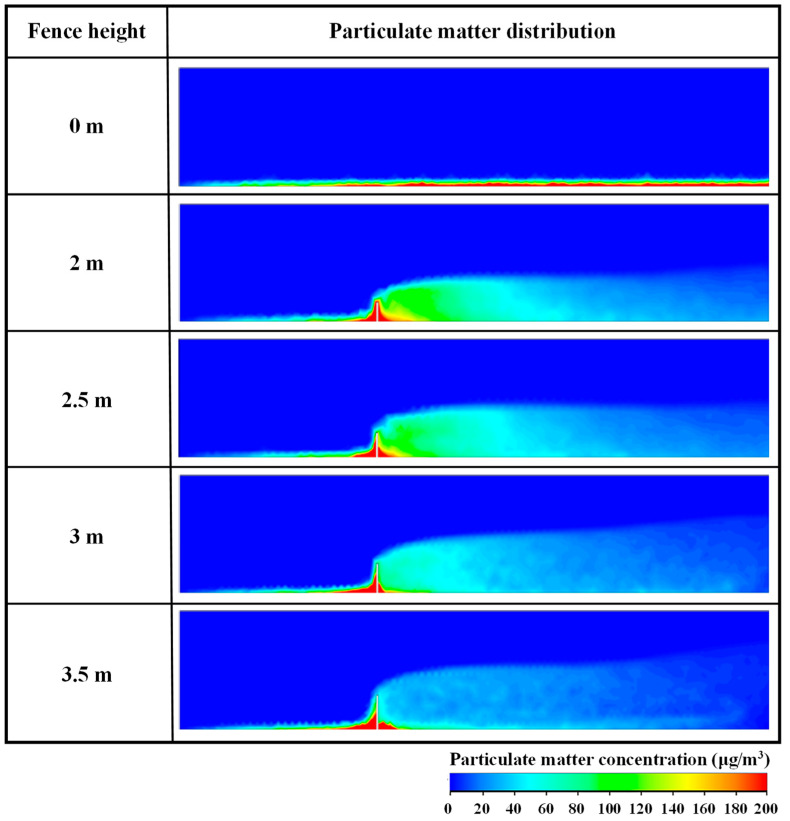
Mass concentration distribution of particles at different enclosure heights on monitoring section S when initial horizontal wind speed is 3 m/s.

**Figure 6 ijerph-20-04361-f006:**
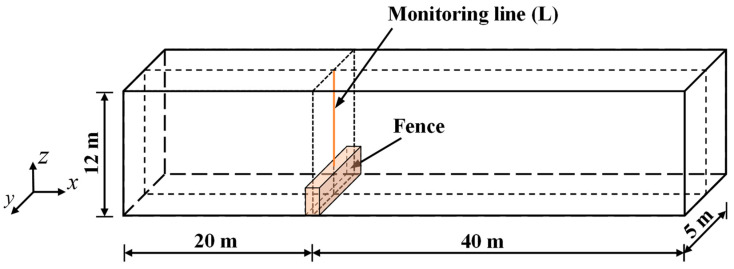
NPSD particle diffusion height in monitoring zone L.

**Figure 7 ijerph-20-04361-f007:**
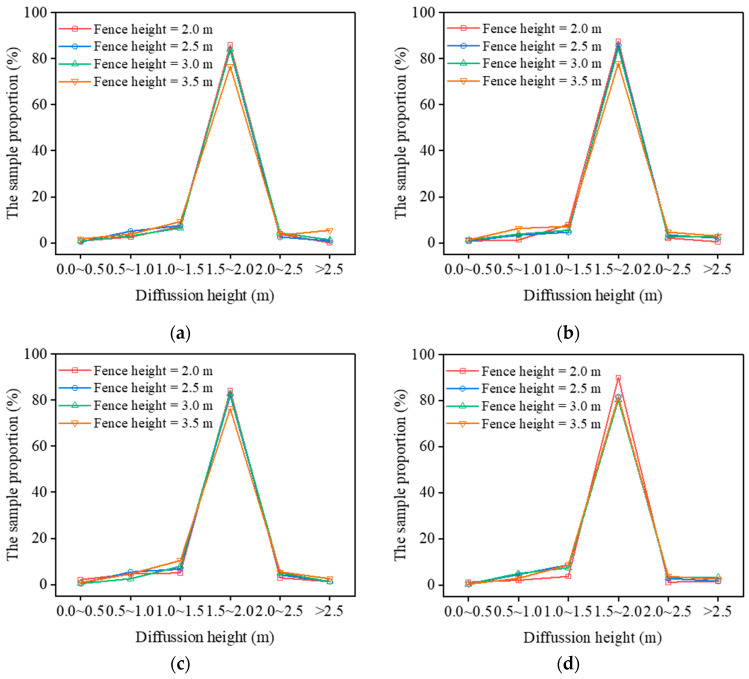
Distribution of diffusion height sample points of NPSD particles above enclosure under different enclosure heights and wind speeds. (**a**) Wind speed *v* = 1 m/s; (**b**) *v* = 2 m/s; (**c**) *v* = 3 m/s; (**d**) *v* = 4 m/s; and (**e**) *v* = 5 m/s.

**Figure 8 ijerph-20-04361-f008:**
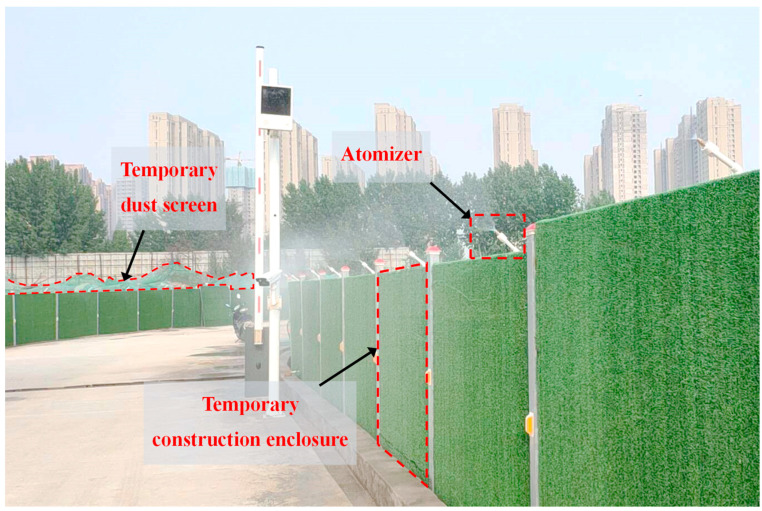
Temporary construction enclosure, dust screen, and atomizer.

## Data Availability

Not applicable.

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
