# Peer review of "Numerical Simulation Study on Spatial Diffusion Behavior of Non-Point Source Fugitive Dust under Different Enclosure Heights"

_ijerph, 2023, doi:10.3390/ijerph20054361_

Round 1

Reviewer 1 Report

In this paper, the authors used numerical simulation methods to analyze the effect of enclosure heights on the particulate diffusion generated from construction. With the analysis results, the authors conclude possible solutions to reduce the particulate emission in residential areas. The methods were clearly described and results are interesting to readers. Thus, it is acceptable with minor revisions. The comments are below:  

1. please pay attention to the grammar errors, for example the usage of passive voice, past tense.

2. In figure 1, what is the height of fence between the construction area and residential area? Even though it’s stated later, please add the description of the fence height will be changed and tested.  In addition, is the mesh evenly distributed in the model? Please provide a zoomed view around the fence area.

3. Please add the reference for the average wind speed in Zhengzhou in line 216.

4. What is the particulate emission standard mentioned in line 331?

5. In 4.4, the results analyzed the percentage of particulates that can reach a specific region. Could it be better if the y axis in figure 7 uses percentage instead of the number of samples? In addition, what’s the difference between figure 7’s (a) to (e)? Please provide the description in the captions or figure titles.

Reviewer 2 Report

What I most concerned is the validation of the simulated results. And the detailed comments were listed in the followed text:

1.      I noticed that the method you employed is a gas-solid two-phase flow which is widely used. And what was the difference between you research and the other studies? In other words, what were the highlights in your study?

2.      Table 1: How did you get the value ranges of these parameters? Please give the detailed explanations.

3.      Section 4.2: Where was section 4.1?

4.      Section 4: I noticed the simulated results were very nice. But, how did the accuracy of you simulated results reach? And, did you validate you results by measured data? If you did not test or validate you simulated results, how can you ensure your simulations could match the actual dust diffussion?

Reviewer 3 Report

Reviewer

General comments:

The manuscript is the result of several experiments through numerical simulation of non-point source particle matter diffusion produced during urban constructions, which represents a thread to air quality in populated regions such as nearby residential areas. The objective of this manuscript was to assess the nature of this diffusion and its interaction with different height enclosures in order to find the best measures to control non-point particle matter diffusion. This work is very interesting and well presented, therefore I recommend its publication. Some clarifications are needed since I believe some readers might have doubts in specific subjects, but overall, the manuscript is in very good state. The introduction is very well written, informative and contextual, with understandable english. Figures are clear and self-explanatory. In addition to the numerical simulation results, authors suggest control measures for non-point source dust beside the enclosure.  

Specific comments:

Line 43 Avoid using pronouns as much as possible.

Line 74 I believe there is a mistake in punctuation here. 

Line 127 This is a little bit confusing. Every CFD simulation should obey the law of conservation of mass and momentum, whether heat transfer is involved or not. Maybe heat transfer is not relevant for this specific simulation, and that would be understood.

Line 130 Since the mass change rate of continuous fluid is being represented in a fixed fluid region it should be used the partial differential equation form of the continuing equation using the term  according to Anderson (1995).

Anderson, J. D. (1995). Computational Fluid Dynamics: The Basics with Applications. McGraw-Hill.

Equation 3. Same as line 130, it should be  according to the conservative form of Navier Stokes equations (Anderson, 1995) for a fixed fluid region model.

Line 174 I think this is quantity instead of quality.

Line 205 Outlet velocity is also not relevant for this work.  

Line 182 I think it is important to mention both spatial and temporal resolution of the simulation.

Line 222 Is there a way to validate some of the findings of these experiments?

Line 255 I suggest to end and start the next idea with a period after “dust particles”.

Figure 3. Vectors are so small in the 3 areas highlighted in every experiment that is very hard to see the vortex next to the enclosure. The vortex must be there by nature, but if it is relevant to mention, since is the main mechanism that accumulates PM next to the enclosure as Figure 4 shows, then vector magnitude reference should be modified in order to highlight this feature. 

Also, the wind speed increases not only above the enclosure but before, probably because of the divergence caused by the increasing since this is apparently proportional to the enclosure height. This an important subject since it could be also affecting the amount of PM dragged by aerodynamical entrainment and be a counter effect.

Line 398 It is important to first explicitly justify why the enclosure height must exceed 3 m, since Figure 5 may give the impression that dust might be actually reaching farther distance into the residential areas even if the concentration is lower, also considering that the recirculation caused in leeward because of the vortex is larger. I believe it needs to be emphasized from the section 4.3 that increasing diffusion is actually a benefit, since it reduces concentration as mentioned in point 2 of conclusions.

I noted that the term “non-source point dust” appears more than 40 times through the manuscript, I recommend abbreviate it.

Also, I have my doubts about what the author is implying about “large-scale numerical simulation” since this might be relative. For some people large scale could be the simulation of a windfarm.
